# Multilingual Large Language Models Are Not (Yet) Code-Switchers

**Ruochen Zhang**[*1]    **Samuel Cahyawijaya**[*2]
**Jan Christian Blaise Cruz**[*3]    **Genta Indra Winata**[*4]    **Alham Fikri Aji**[*5]
[1]Brown University   [2]HKUST
[3]Samsung R&D Institute Philippines   [4]Bloomberg   [5]MBZUAI
ruochen_zhang@brown.edu   scahyawijaya@connect.ust.hk
jcb.cruz@samsung.com   gwinata@bloomberg.net   alham.fikri@mbzuai.ac.ae

## Abstract

Multilingual Large Language Models (LLMs) have recently shown great capabilities in a wide range of tasks, exhibiting state-of-the-art performance through zero-shot or few-shot prompting methods. While there have been extensive studies on their abilities in monolingual tasks, the investigation of their potential in the context of code-switching (CSW), the practice of alternating languages within an utterance, remains relatively uncharted. In this paper, we provide a comprehensive empirical analysis of various multilingual LLMs, benchmarking their performance across four tasks: sentiment analysis, machine translation, summarization and word-level language identification. Our results indicate that despite multilingual LLMs exhibiting promising outcomes in certain tasks using zero or few-shot prompting, they still underperform in comparison to fine-tuned models of much smaller scales. We argue that current "multilingualism" in LLMs does not inherently imply proficiency with code-switching texts, calling for future research to bridge this discrepancy.

## 1 Introduction

Large Language Models (LLMs) have shown their potential in the context of zero-shot and few-shot prompting (Brown et al., 2020; Kojima et al., 2022; Wei et al., 2022; Longpre et al., 2023). The successes of these LLMs have also been effective in multilingual settings (Lin et al., 2021; Winata et al., 2021b; Scao et al., 2022) where models are specifically trained to learn individual languages, proven to be highly beneficial for monolingual tasks. However, in multilingual communities, people do not confine themselves to speaking only a single language; instead, they use two or more languages interchangeably during a conversation - a phenomenon known as code-switching (CSW) (Poplack, 1980, 2001). It allows individuals to communicate cultural-specific concepts more effectively, signaling their group identity and reinforcing their social connection (Doğruöz et al., 2021). Yet, existing multilingual LLMs are not specifically trained with objectives for managing CSW scenarios. Hence, assessing the capabilities of the current multilingual LLMs in processing CSW texts is essential to the development of multilingual language models that are fully compatible with code-switching.

The main challenge of developing multilingual LLMs optimized for code-switching lies in data scarcity. Given the highly colloquial characteristic of code-switching (Winata et al., 2022b), existing resources dedicated to CSW are rare and collection at large-scale requires considerable annotation efforts. To mitigate such deficiency, Yong et al. (2023) investigate the possibility of employing multilingual LLMs to generate high-quality synthetic CSW texts. The study revealed that, hosted LLMs, such as InstructGPT (Ouyang et al., 2022) and ChatGPT[1] outperform public models like BLOOMZ (Muennighoff et al., 2022) and Flan-T5-XXL (Chung et al., 2022) in generating natural-sounding CSW texts. However, the quality of the generated text by these hosted LLMs is mostly confined to Singlish and significantly declines when prompted for other languages. Despite the preliminary promising results, the generation of high-quality CSW texts still remains challenging. This observation motivates us to probe from a different perspective - *Can existing multilingual LLMs effectively understand CSW?*

There have been previous studies on evaluating multilingual LMs in CSW scenarios (Tan and Joty, 2021; Adilazuarda et al., 2022), where code-switched texts are simulated by replacing words from parallel corpora. Winata et al. (2021a) also assesses models' effectiveness by experimenting with word embeddings constructed from different methods. These works are mainly built upon small

---

[*]Equal contribution.

[1]https://chat.openai.com/

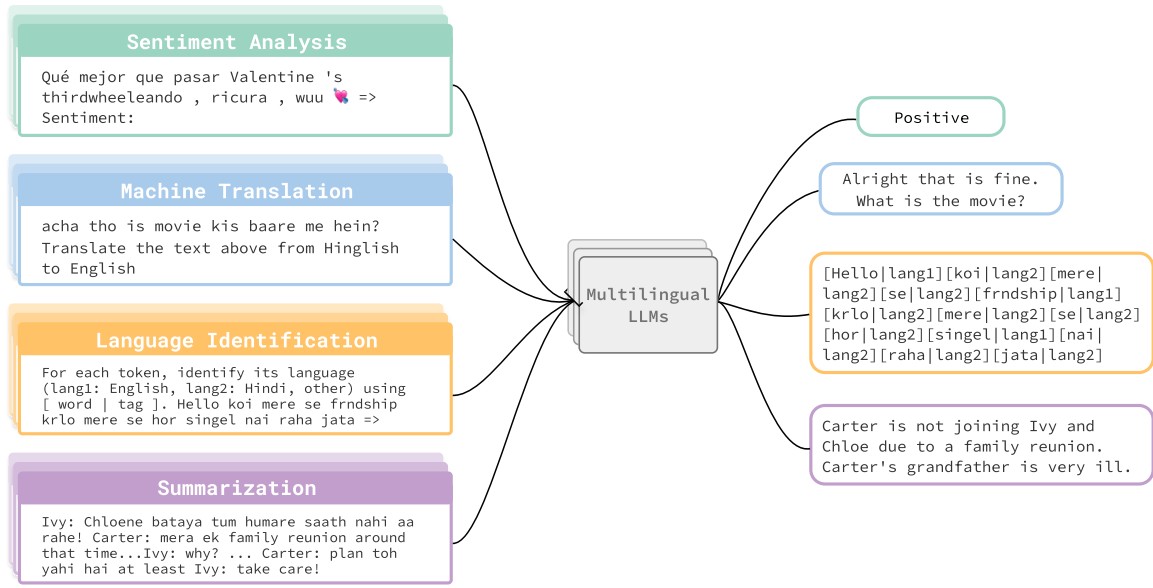

Figure 1: Illustration of tasks included in our benchmark study.

BERT-based models and are restricted by either the number of languages or tasks investigated. Given the recent success of prompting methods on multilingual LLMs and the effects of scaling, this paper presents a more comprehensive empirical analysis of models' code-switching abilities, including a variety of languages, task types, model architectures, model sizes and prompting methods.

Our results suggest that the scaling law is applicable to multilingual LLMs across diverse CSW tasks and model architectures. However, fine-tuned smaller-scale models substantially outperform the largest multilingual LLM with prompting methods. In addition, while hosted LLMs achieve scores comparable to our fine-tuned models, such performance remains uninterpretable due to their closedness. We argue that existing multilingual LLMs exhibit limited proficiency in code-switching contexts, highlighting future research opportunities to transform them into true polyglots.

## 2 Experimental Setup

### 2.1 Datasets

We explore four code-switching task categories: sentiment analysis (SA), machine translation (MT), summarization (SUM), and word-level language identification (LID). The description of each task is as follows:

**Sentiment Analysis** We use sentiment analysis datasets of three different language pairs: Sentimix Spanish-English (Aguilar et al., 2020), MixSenti-

ment Malayalam (Chakravarthi et al., 2020a), and MixSentiment Tamil (Chakravarthi et al., 2020b). Besides the common positive and negative labels, these datasets also contain extra labels like neutral or other. However, occurrences of those labels are very scarce. Hence, to normalize datasets from different sources, we simplify the data by filtering out examples outside positive or negative labels. The dataset sizes, broken down into train/validation/test, are as follows: 8,831/8,831/1,342 pairs for the Spanish-English subset, 2,541/275/703 for Malayalam, and 9,075/1,022/2,499 for the Tamil subset.

**Machine Translation** We use the code-switched datasets from MixMT 2022 shared task (Srivastava and Singh, 2022) that contains Hinglish-English sentence pairs (8,060 pairs in the training split, 942 pairs in validation and 960 pairs in the test split).

**Summarization** We use code-switched summarization dataset Gupshup (Mehnaz et al., 2021), which is derived from SAMSum (Gliwa et al., 2019) via crowdsourcing translation. In our experiment, We focus on Hinglish to English, as evaluating code-switched summary systematically with existing auto metrics has shown to be challenging for multilingual LLMs (Zhang and Eickhoff, 2023). The dataset contains 5,831 source-target pairs for training, with 500 pairs each for validation and testing.

**Word-level LID** We use English-Hindi and Modern Standard Arabic (MSA) - Egyptian Arabic (EA) subsets from the Language Identification task in

| Model | Model Type | | | Model Sizes | Datasets | # Languages | LLM Objectives |
|---|---|---|---|---|---|---|---|
| | Enc-Only | Dec-Only | Enc-Dec | | | | |
| XLM-R (Conneau et al., 2020) | ✓ | | | 250M, 560M | CommonCrawl | 100 | MLM |
| mBERT (Devlin et al., 2019) | ✓ | | | 178M | Wikipedia | 104 | MLM |
| mDeBERTa v3 (He et al., 2021) | ✓ | | | 278M | CC100 | 100 | RTD w/ GDES |
| mBART-50 (Tang et al., 2020) | | | ✓ | 611M | CC25, ML50 | 50 | Denoising w/ CLM |
| M2M100 (Fan et al., 2020) | | | ✓ | 418M, 1.2B | CCMatrix, CCAligned | 100 | CLM |
| XGLM (Lin et al., 2021) | | ✓ | | 564M, 1.7B, 2.9B, 4.5B, 7.5B | CommonCrawl | 30 | CLM |
| BLOOMZ (Muennighoff et al., 2022) | | ✓ | | 560M, 1.1B, 1.7B, 3B, 7.1B | ROOTS, xP3 | 46 | Instruction Tuned |
| mT0 (Muennighoff et al., 2022) | | | ✓ | 300M, 580M, 1.2B, 3.7B, 13B | mC4, xP3 | ∼120 | Instruction Tuned |
| ChatGPT (Bang et al., 2023) | | ✓ | | - | - | - | RLHF |

Table 1: Comparison of different model variants studied in this paper.

the LinCE benchmark (Aguilar et al., 2020). In this task, the system is tasked with classifying the language of each word in a sentence into one of the three classes, lang1, lang2, or other. lang1 and lang2 are English and Hindi, or MSA and EA, respectively. The English-Hindi subset contains 4,832 training examples and 744 validation examples. For the MSA-EA subset, it contains 8,464 examples for training and 1,116 for validation. Our results are reported on the validation set as the test set is unavailable publicly.

## 2.2 Models

**Zero-shot and Few-shot Models** For zero-shot and few-shot prompting, we explore various multilingual generative LLMs of different pretraining processes and architectures, including BLOOMZ, mT0 (Muennighoff et al., 2022) and XGLM (Lin et al., 2021). We explore all model sizes except for BLOOMZ 175B due to resource limitations. We also include ChatGPT into our analysis and specifically GPT-3.5$_{\text{turbo}}$ is used. We explore 0, 1, 3, and 5-shot on each model with 5 diverse prompt templates. Details for each prompt can be seen in Appendix C.

For the SA task, we compute the probability of the model to generate each label as the next immediate continual generation, and then we pick the label resulting in the highest probability for the whole sequence. For MT, SUM and LID, we perform standard text generation. However, for LID, we expect the generated text to follow a predefined format where each [token, language tag] pair is represented as [ token | tag ]. We parse the generation using a dynamic programming algorithm introduced in Paolini et al. (2021) to extract the valid [token, language tag] pairs for evaluation.

**Fine-tuning Models** In addition to zero-shot prompting models and few-shot in-context learning, we also experiment with fine-tuning as a bench-

mark against prompting. For SA and word-level LID tasks, we fine-tune four models, namely, base and large variants of XLM-RoBERTa (Conneau et al., 2020), mBERT (Devlin et al., 2019), and mDeBERTa v3 (He et al., 2021).

For MT, we fine-tune eight models in total. These include small, base, and large variants of mT0 (Muennighoff et al., 2022); 418M and 1.2B variants of M2M100 (Fan et al., 2020); and standard, one-to-many, and many-to-many variants of mBART-50 (Liu et al., 2020; Tang et al., 2020)[2]

For SUM, we follow the same setup used in MT, except we only fine-tune the three previously mentioned mT0 models and only the standard mBART-50 as the one-to-many and many-to-many variants are specifically for translation only.

Across all the tasks, we fine-tune the selected models on all the available training instances. Table 1 shows a full overview and comparison of the models investigated in this study and details for training setups for all tasks can be found in Appendix A.

## 3 Results and Discussion

**Overall Results** Figure 2 presents the results of various multilingual LLMs for the four CSW tasks.[3] In general, we observe a scaling pattern when prompting multilingual LLMs across tasks. Nevertheless, the performance of these models significantly falls short when compared to that of substantially smaller fine-tuned models. Therefore, adopting a fine-tuned model is a more practical approach for dealing with CSW tasks, especially in scenarios with constrained computational resources. For ChatGPT, it demonstrates comparable performance to fine-tuned models across all tasks

---

[2]Due to space constraint, we show a selection of all fine-tuned models in Table 2. For the full results, please refer to Appendix B.

[3]Note that the results for SA, MT, SUM are derived from zero-shot prompting while LID results are based on 5-shot.

| Sentiment Analysis | | | | Machine Translation | | | Summarization | | Language Identification | | |
|---|---|---|---|---|---|---|---|---|---|---|---|
| **Model** | **F1** | | | **Model** | **BLEU** | | **Model** | **RL** | **Model** | **F1** | |
| | Mal-Eng | Spa-Eng | Tam-Eng | | Hng$^\beta$→Eng | Eng→Hng$^\beta$ | | Hng$^\delta$→Eng | | Hin-Eng | MSA-EA |
| **Finetuning** | | | | **Finetuning** | | | **Finetuning** | | **Finetuning** | | |
| XLMR$_{278M}$ | 77.08 | 77.14 | 68.12 | M2M100$_{418M}$ | 28.53 | 12.40 | mT0$_{p,300M}{}^\delta$ | 29.83 | XLMR$_{278M}$ | 82.44 | 72.58 |
| XLMR$_{560M}$ | **79.94** | 78.81 | **68.28** | mBART50$_{610M}{}^\gamma$ | 29.53 | 13.38 | mT0$_{p,580M}{}^\delta$ | 37.44 | XLMR$_{560M}$ | **86.65** | **79.79** |
| mBERT$_{178M}$ | 78.21 | 70.02 | 65.19 | mT0$_{p,580M}{}^\delta$ | 25.47 | 12.28 | mT0$_{p,1.2B}{}^\delta$ | **40.12** | mBERT$_{178M}$ | 81.99 | 68.02 |
| mDeBERTa$_{278M}$ | 44.56 | **88.17** | 45.56 | mT0$_{p,1.2B}{}^\delta$ | **31.88** | **13.90** | mBART50$_{610M}$ | 39.03 | mDeBERTa$_{278M}$ | 85.41 | 68.02 |
| **0-shot Prompting** | | | | **0-shot Prompting** | | | **0-shot Prompting** | | **5-shot Prompting** | | |
| mT0$_{300M}$ | 36.79 | 48.44 | 42.26 | mT0$_{300M}$ | 2.74 | 1.60 | mT0$_{300M}$ | 16.00 | mT0$_{300M}$ | 2.13 | 0.90 |
| mT0$_{580M}$ | 44.60 | 56.01 | 47.62 | mT0$_{580M}$ | 6.42 | 2.37 | mT0$_{580M}$ | 20.16 | mT0$_{580M}$ | 0.30 | 0.00 |
| mT0$_{1.2B}$ | 55.62 | 67.63 | 53.88 | mT0$_{1.2B}$ | 10.64 | 1.88 | mT0$_{1.2B}$ | 23.63 | mT0$_{1.2B}$ | 0.22 | 0.27 |
| mT0$_{3.7B}$ | 35.27 | 59.28 | 38.55 | mT0$_{3.7B}$ | 12.78 | 2.08 | mT0$_{3.7B}$ | 27.40 | mT0$_{3.7B}$ | 0.19 | 1.49 |
| mT0$_{13B}$ | 49.97 | 65.26 | 50.76 | mT0$_{13B}$ | 19.28 | 1.66 | mT0$_{13B}$ | 30.67 | mT0$_{13B}$ | 7.51 | 5.07 |
| BLOOMZ$_{560M}$ | 59.64 | 72.79 | 55.30 | BLOOMZ$_{560M}$ | 2.24 | 1.37 | BLOOMZ$_{560M}$ | 14.22 | BLOOMZ$_{560M}$ | 5.38 | 2.08 |
| BLOOMZ$_{1.1B}$ | 50.64 | 70.89 | 53.27 | BLOOMZ$_{1.1B}$ | 2.79 | 1.73 | BLOOMZ$_{1.1B}$ | 16.45 | BLOOMZ$_{1.1B}$ | 16.31 | 10.56 |
| BLOOMZ$_{1.7B}$ | 47.83 | 73.20 | 50.15 | BLOOMZ$_{1.7B}$ | 2.62 | 2.62 | BLOOMZ$_{1.7B}$ | 16.85 | BLOOMZ$_{1.7B}$ | 13.04 | 3.37 |
| BLOOMZ$_{3B}$ | 56.84 | 72.85 | 53.41 | BLOOMZ$_{3B}$ | 3.13 | 2.86 | BLOOMZ$_{3B}$ | 20.97 | BLOOMZ$_{3B}$ | 19.61 | 17.47 |
| BLOOMZ$_{7B}$ | 64.21 | 74.61 | 59.43 | BLOOMZ$_{7B}$ | 3.67 | 1.88 | BLOOMZ$_{7B}$ | 17.01 | BLOOMZ$_{7B}$ | 19.58 | 9.26 |
| XGLM$_{564M}$ | 52.18 | 64.16 | 52.66 | XGLM$_{564M}$ | 0.45 | 0.28 | XGLM$_{564M}$ | 4.29 | XGLM$_{564M}$ | 6.65 | 1.61 |
| XGLM$_{1.7B}$ | 50.83 | 65.01 | 50.55 | XGLM$_{1.7B}$ | 0.79 | 0.43 | XGLM$_{1.7B}$ | 5.42 | XGLM$_{1.7B}$ | 5.90 | 6.27 |
| XGLM$_{2.9B}$ | 60.15 | 64.78 | 56.43 | XGLM$_{2.9B}$ | 1.34 | 0.69 | XGLM$_{2.9B}$ | 5.75 | XGLM$_{2.9B}$ | 17.64 | 10.75 |
| XGLM$_{4.5B}$ | 62.32 | 70.34 | 56.94 | XGLM$_{4.5B}$ | 2.13 | 0.47 | XGLM$_{4.5B}$ | 4.73 | XGLM$_{4.5B}$ | 19.35 | 20.51 |
| XGLM$_{7.5B}$ | 60.93 | 68.52 | 56.04 | XGLM$_{7.5B}$ | 1.43 | 0.39 | XGLM$_{7.5B}$ | 5.92 | XGLM$_{7.5B}$ | 16.91 | 18.91 |
| GPT-3.5$_{turbo}$ | **65.92** | **75.64** | **63.15** | GPT-3.5$_{turbo}$ | **27.64** | **4.32** | GPT-3.5$_{turbo}$ | 25.07 | GPT-3.5$_{turbo}{}^\alpha$ | **80.19** | **71.41** |

$^\alpha$ Due to budget limitations, the results presented in GPT-3.5$_{turbo}$ are based on 1-shot prompting instead of 5-shot.

$^{\beta,\gamma,\delta}$ Hng refers to Hinglish, a mix of Hindi and English. mBART50 refers to the many-to-many variant. mT0$_p$ refers to the fine-tuned mT0 with prompt templates.

Table 2: Code-switching benchmark results for finetuned and prompting models. We report the 0-shot performance for the sentiment analysis, machine translation and summarization tasks; and 5-shot performance for the word-level language identification task.

and datasets, except for the English to Hinglish MT task. This exception may stem from the challenges in generating code-switched texts as outlined in previous research (Yong et al., 2023; Zhang and Eickhoff, 2023). For the remaining tasks, ChatGPT notably outperforms publicly available multilingual LLMs. Such discrepancy may be attributed to the RLHF objective in its pretraining process, although a comprehensive analysis is hindered by its proprietary nature.

## 3.1 Sentiment Analysis Results

Figure 5 shows a detailed breakdown for each of the three language datasets in the SA task. The results from fine-tuned models mainly reside in the top-left corner across all three datasets, highlighting their superior performance with considerably smaller sizes. Scaling BLOOMZ and XGLM yield small improvements, however, scores from mT0 fluctuate around 50 F1 when varying sizes. It's worth noting that the majority-class baseline of these three datasets has an average F1 score of 46. Considering the instability observed during the scaling-up process, mT0 struggles to understand the sentiment when presented in CSW texts.

## 3.2 Machine Translation Results

As shown in Figure 2 and Table 2, when the source is Hinglish and target English, the performance gap between prompting and fine-tuning in MT is much more apparent, with the best prompted LLM mT0-XXL achieving no more than 20 BLEU while all the fine-tuned models achieved between 25-32 BLEU score. In contrast to SA, we notice especially visible improvement during scaling up encoder-decoder style models such as mT0, while decoder-only models such as BLOOMZ and XGLM have minimal improvements given their overall poor performance.

We then compare the difference in LLM scaling between translation tasks with code-switched sources and monolingual ones[4]. Figure 3 shows the scaling trajectory of LLMs for both Hindi → English and Hinglish → English translation direction; Table 3 presents the regression coefficient ($\beta$) in these two scenarios. A large coefficient indicates scaling has more noticeable impacts. We can observe that the influence of scaling is more apparent in monolingual sources than in the code-switched

[4]Monolingual experiments are conducted on WMT 2014 Hindi-English dataset (Bojar et al., 2014).

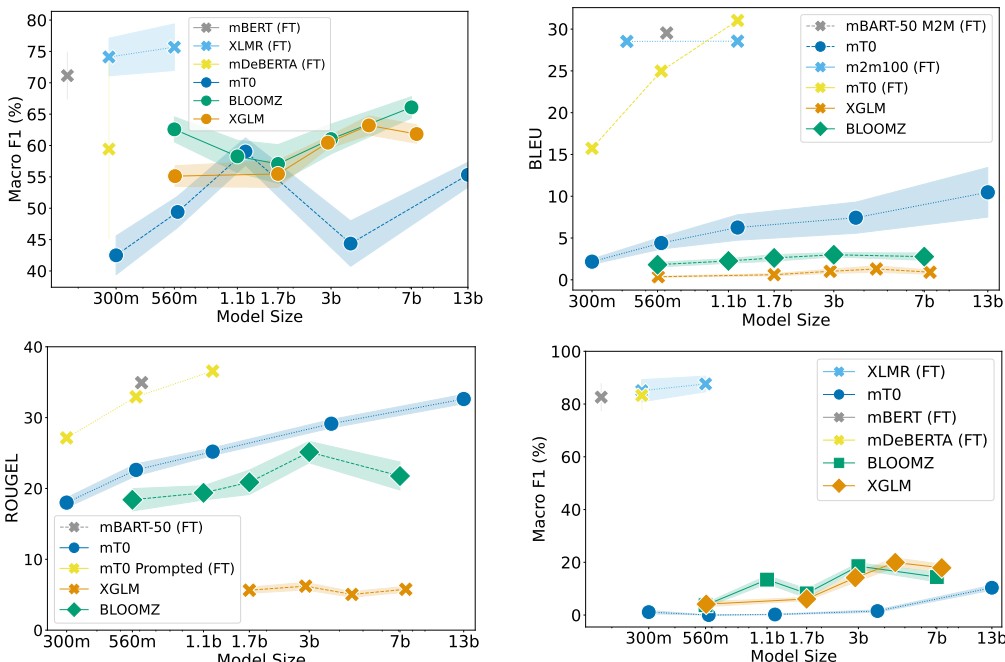

Figure 2: Evaluation results of fine-tuning and prompting LLMs of different scales on various CSW tasks. **(top left)** F1-score on the sentiment analysis task, **(top right)** BLEU score on the machine translation task, **(bottom left)** ROUGE-L on the summarization task, and **(bottom right)** F1-score on the word-level language identification task. (FT) means results are from fine-tuned models.

setup. This pattern could potentially result from the limited pretraining samples for Hinglish code-switched data, leading to a sub-optimal scaling performance.

When models are tasked with translating the source into CSW text, a substantial performance decline is observed for both fine-tuned and prompted models. We notice that while the larger mT0 models are capable of producing English translations in a zero-shot manner, they struggle to generate CSW texts as seen in previous work (Yong et al., 2023). Upon looking at the output, mT0 simply outputs in English, even in few-shot settings in which it has seen some other Hinglish examples.

### 3.3 Summarization Results

Figure 2 shows the fine-tuning and zero-shot prompting result on the summarization task. Similarly, we see that fine-tuned models outperform the zero-shot approach. Similar to MT, mT0 yields the overall best performance and shows positive scaling law.

To disentangle CSW from the equation, we evaluate the LLM's performance on the same Gupshup dataset, but with English input rather than Hinglish input. The evaluation set is parallel to each other. Interestingly, from Figure 3 and Table 3, we see a

similar scaling impact whether the input is monolingual or in code-switch. However, the models are consistently better if the input is in English.

### 3.4 Language Identification Results

Our observation of fine-tuned models in the LID task is similar to the MT task: they outperform prompting methods on multilingual LLMs by a significant margin. In Table 2, we report 5-shots instead of 0-shot prompting results for LID tasks as 0-shot results are all 0 for both language datasets and across all models. The multilingual LLMs are not able to understand the natural language instruction that requires them to generate outputs in a specific format like [ token | tag ] word by word. When prepending more in-context examples in the instruction, we observe slight performance improvements across different models. For results on few-shot experiments for LID, please refer to Section 3.5.

### 3.5 Few-Shot Results

Compared to zero-shot inference, few-shot learning has been shown to boost performance as discussed in previous works(Brown et al., 2020; Liu et al., 2021). However, in CSW settings, we observe different effects of adding more in-context examples

between tasks. In Figure 4, we notice a decrease in metrics from 0-shot to 1-shot for SA and SUM, suggesting that in-context examples do not contribute to or even degrade models' performance. We suspect that models have seen these tasks in a monolingual fashion during pretraining, and thus are able to understand instructions well in a zero-shot setting. Instead, models may consider CSW examples as low-quality texts, thus confusing the generation process. For MT, we observe negligible change in the models' performances with an increasing number of examples. Notably, instead of translating sentences to Hinglish as instructed, models could only repeat the original English sentences. For instance, when provided with 5 in-context examples, mT0$_{13B}$ is instructed to "Translate the following text from English to Hinglish. Text: hello there, I have not seen this movie so im going to take a minute to look it over :) Translation:". It generates "hello there, I have not seen this movie so I going to take time to look it over:)." instead of the expected "hello yar, mein is movie ko nahi dekha hoon tho, tho mein thode der ke liye isko dekh loonga". Similar issues are also observed with BLOOMZ. We hypothesize that models may not fully comprehend the nuances of 'Hinglish' within the given instruction, which could account for their relatively uniform performance across varying shot numbers.

On the contrary, more in-context examples benefit the LID task. As no models are pre-trained on the sequence tagging task, the natural instruction entailing the specific generation format is new to the LLMs. Therefore, in our experiments, most models perform best when given 5 learning examples. Additionally, though we observe scaling law patterns in 5-shot settings as shown in Figure 6, for the best-performing billion-parameter models, we still consistently observe their inability to adhere to the format laid out in the instructions. They often fail to replicate the exact words required for sentence tagging or predict multiple tokens within a single bracket pair. For example, in a 5-shot setting, when asked to label the sentence "we the fans luv you , sirji", BLOOMZ$_{7b}$ wrongly generates "[ we the fans | lang1 ] [ you | lang1 ] [ sirji | lang1 ] [ , | other ]", unable to put individual words in the brackets and omitting some words from the original sentence. Given the constraint of limited input length, which restricts the number of in-context examples models can learn from, their uncontrollable

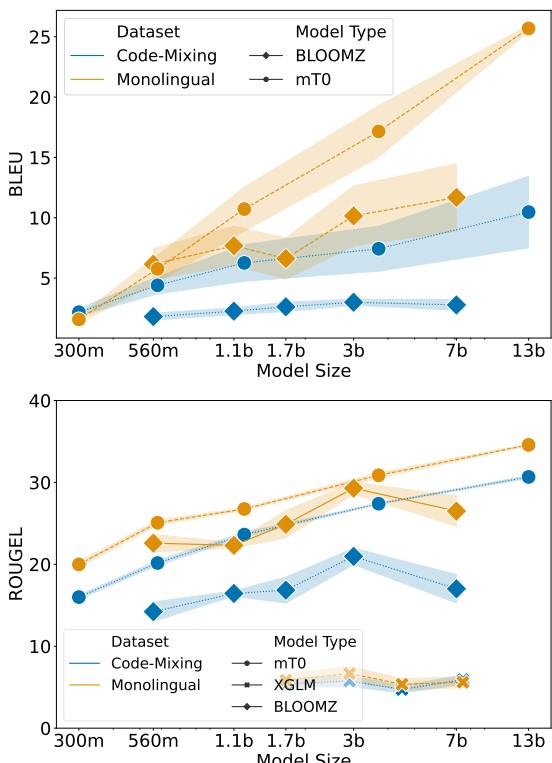

Figure 3: Performance comparison on **(top)** Hindi→English vs Hinglish→English translation and **(bottom)** Hinglish→English vs English→English summarization.

generation still results in a significant performance gap when compared to fine-tuning smaller models (∼20 F1 vs. ∼80 F1).

## 3.6 Benchmarking ChatGPT

Given recent developments in general-purpose, instruction-following LLMs like ChatGPT, with impressive zero-shot abilities across tasks, we also benchmark ChatGPT's performance in our CSW task. Limited by the budget, we only explore zero-shot performance for SA, MT and SUM given their easy scopes, and 1-shot performance for LID due to the specific output format requirements. Since we can't access ChatGPT's output probability distribution, we instruct ChatGPT to return only the exact string label and calculate F1 scores using exact string matching for SA.

ChatGPT achieves somewhat comparable performance to finetuning models and significantly outperforms other public multilingual LLMs in most of the tasks. Especially for LID, it shows strong capabilities in following difficult instructions with only one example. The only exception is on the English→Hinglish MT tasks, where its zero-shot

| Model | Code-Switched | | Monolingual | |
|---|---|---|---|---|
| | $\beta$ | $\alpha$ | $\beta$ | $\alpha$ |
| Machine Translation | | | | |
| mT0 | 1.057 | 6.403 | 1.626 | 6.075 |
| BLOOMZ | 0.192 | 2.373 | 0.824 | 6.240 |
| Summarization | | | | |
| mT0 | 0.712 | 5.471 | 0.738 | 9.228 |
| BLOOMZ | 0.312 | 3.507 | 0.644 | 8.637 |
| XGLM | 0.029 | 0.444 | 0.012 | 0.883 |

Table 3: Regression slope ($\beta$) and intercept ($\alpha$) of scaling mT0 and BLOOMZ on monolingual/code-switched machine translation and summarization task.

performance is only slightly better than other public LLMs. We hypothesize mainly two reasons behind the difficulty in generating CSW texts: 1) as alluded to in the previous section, CSW texts can be perceived as noises given tasks and pretraining processes are designed in a monolingual fashion; 2) LLMs may have a lack of sufficient representation for CSW text structure. In our analysis, LLMs perform much better in SA tasks as they could pick up cues from individual works instead of paying attention to language "structure" when tasked with text generation.

Lastly, while ChatGPT delivers promising results without any fine-tuning, the lack of complete transparency on its pretraining datasets, model architecture, and training details obstructs a better understanding of its performance. This presents roadblocks to future improvements in code-switching proficiency for public multilingual LLMs.

## 4 Implications for Future LLMs

In this section, we walk through various implications of our work and provide recommendations for enabling better CSW ability in LLMs. By highlighting this limitation, we compel researchers to consider CSW as a core feature of many people's multilingual repertoire across the world.

**Fairer Data Representation for Code-Switching** Our results in Section 3 show that existing LLMs have similar scaling patterns between monolingual and CSW. However, despite all the models under study having seen each of the languages during pretraining, there is still a performance gap between monolingual and CSW. This suggests that the ability to code-switch is not acquired by LLMs after

pretraining and/or instruction-tuning with multilingual data (Xue et al., 2021; Scao et al., 2022; Muennighoff et al., 2022), indicating the need for adding better data representation for code-switching in the multilingual pretraining and/or instruction-tuning process. Such an approach can be done through manual CSW data collection and/or various data augmentation methods (Tan and Joty, 2021; Adilazuarda et al., 2022; Dhole et al., 2023). Aside from adding more CSW data, one potential solution is to identify and include the code-switching language pairs into consideration of multilingual pretraining and/or instruction-tuning. This allows better resampling strategy (Lample and Conneau, 2019; Aharoni et al., 2019; Conneau et al., 2020; Xue et al., 2021; Tang et al., 2021; Cahyawijaya et al., 2021) for CSW data during the multilingual pretraining and/or instruction-tuning.

**Adaptation and Extension of Code-Switching Optimization Objectives** Existing LLMs are optimized solely with language modeling objectives either for sentence denoising or sentence completion. However, alternative optimization objectives,, such as meta transfer learning (Winata et al., 2020) and additional token/span-level language identification objective (Li et al., 2019), have been demonstrated to effectively enhance CSW performance with minimal performance loss on monolingual tasks in CSW speech processing. By adapting and extending these approaches to NLP, we may be able to equip LLMs with better CSW capability without requiring expensive data collection and annotation. This would be particularly advantageous for LLMs, especially in applications where CSW is prevalent within the multilingual community.

**Towards More Inclusive Language Technology** In light of the fact that LLMs are the driving force behind the progress of various NLP technologies (Thoppilan et al., 2022; SambaNova Systems, 2023; Pratap et al., 2023), we emphasize the importance of incorporating code-switched capabilities in LLMs to promote inclusivity and diversity in language technology, particularly for multilingual speakers who frequently engage in code-switching in their daily lives. By enabling NLP technology to reflect the language-mixing patterns of users, people can communicate in ways that are more comfortable and authentic to their linguistic identities, eliminating the need for people to adjust their speech patterns to become legible to machines. It

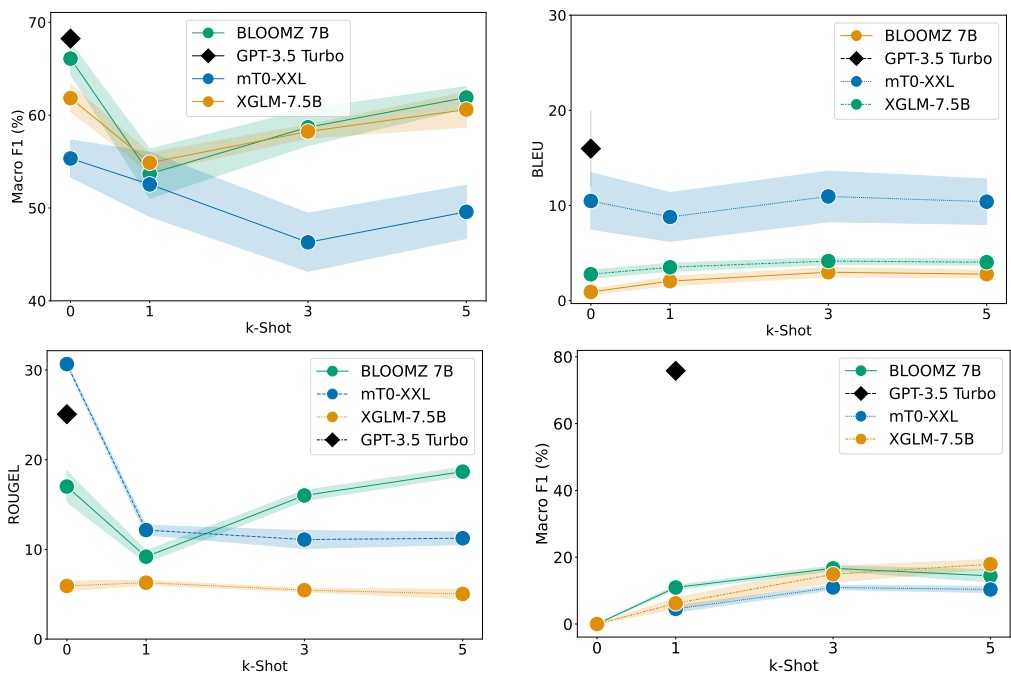

Figure 4: Few-shot evaluation performance for **(top left)** sentiment analysis task, **(top right)** machine translation task, **(bottom left)** summarization task and **(bottom right)** word-level LID task.

would not only mitigate the effects of linguistic profiling (Baugh, 2005; Dingemanse and Liesenfeld, 2022) and hegemonic, Western-centric technological designs but also foster greater trust among users in language technology through naturalistic dialogue interactions. Therefore, we urge the integration of code-switched recognition and generation capabilities in future LLMs.

## 5   Related Work

**Code-Switching**   Code-switching is a common practice observed in multilingual communities where people mix multiple languages within an utterance (Poplack, 2001). While more than half of the world population speaks more than one language, the availability of resources and assessments for code-switching is much more limited compared to the extensive literature on monolingual cases. The key challenges of collecting high-quality code-switching data lie in the colloquial nature of the practice and the language proficiency required for accurate annotation (Winata et al., 2022b). The recent advances of "multilingual" large language models compel one to explore whether these models are proficient in code-switching contexts like a true polyglot. Previous research (Winata et al., 2021a) has studied the code-switching capabilities of language models in NER and POS-tagging tasks, however, the work is limited to using only differ-

ent word embeddings and encoder-only models. In this paper, we expand on previous works and provide a detailed analysis of more model variations, task objectives and downstream applications of diverse language pairs adopted from existing CSW benchmarks like LinCE (Aguilar et al., 2020) and GlueCOS (Khanuja et al., 2020).

**Multilingual Large Language Models**   Models like mBERT (Devlin et al., 2019) and XLM-R (Conneau et al., 2020) have become the go-to multilingual options for supervised fine-tuning, given their impressive abilities and adaptability to many languages. With the success of large-scale generative models, their capabilities have been enriched with multilingual objectives (Lin et al., 2021; Scao et al., 2022; Muennighoff et al., 2022) through pretraining on large multilingual corpora such ROOTS (Laurençon et al., 2022), mC4 (Raffel et al., 2019) and xP3 (Muennighoff et al., 2022). In addition to excelling in different monolingual and multilingual benchmarks via zero-shot prompting (Sanh et al., 2021; Wei et al., 2021; Kojima et al., 2022; Muennighoff et al., 2022; Bang et al., 2023), research has shown that scaling up model sizes (Cahyawijaya et al., 2023; Kaplan et al., 2020; Fernandes et al., 2023) and incorporating in-context learning examples (Winata et al., 2022a; Tanwar et al., 2023) could help further boost

their performance. Yet, given the scarcity of CSW evaluation resources, how these multilingual LLMs perform in code-switching scenarios still remains questionable. In this paper, we evaluate these models under various settings including fine-tuning, zero-shot prompting, and in-context learning, and provide recommendations for future improvements in code-switching proficiency.

# 6 Conclusion

In this paper, we systematically study multilingual LLMs' capabilities in code-switching tasks along various dimensions, including but not limited to finetuning vs. prompting, task objectives, scaling laws and model architecture. We observe that, despite improvements with larger sizes, existing multilingual LLMs still yield inferior performance compared to fine-tuning smaller models. We argue that multilingual LLMs are not necessarily code-switching compatible. Given that multilingual LLMs are not explicitly trained for code-switching data, we recommend future development should incorporate a more comprehensive evaluation framework that encompasses code-switching texts. Finally, our study is limited to models' performance in sentiment analysis, machine translation, summarization and language identification. We suggest that benchmarking across a broader set of tasks is required. However, the scarcity of high-quality open-source code-switching datasets and the challenges associated with their collection process imply future work should also include constructing code-switching data with more complexity, such as commonsense reasoning.

## Limitations

The scope of code-switching languages in this work is limited to Hindi-English, Standard-Egyptian Arabic, Spanish-English, Tamil-English, and Malayalam-English. It is beneficial to include more languages to demonstrate the generality of our claim. However, a challenge in doing so arises from the lack of available code-switched text datasets. We explore four different NLP downstream tasks. However, similar to the previous point, it would be interesting to cover more tasks. Similarly, the main challenge of expanding into different tasks is the lack of available datasets. We anticipate that future studies will broaden the exploration of code-switching languages and tasks beyond those examined in this research to showcase the generalizability of the findings to other code-switching languages and tasks.

In addition, in this study, we choose multilingual LLMs based on two criteria: 1) they present or advertise themselves as multilingual and 2) their pretraining data contain all the languages featured in our benchmark dataset. Although some recently released LLMs like Llama-2 (Touvron et al., 2023) and Falcon (Penedo et al., 2023) have demonstrated state-of-the-art performance across various other benchmarks, we defer the evaluation of their code-switching capabilities to future research.

Finally, our observations are based on the model sizes allowed by our local compute resources. A more comprehensive analysis can be obtained by experimenting with a wider range of variations, including larger model sizes and more in-context examples given a more generous compute budget.

## Ethical Considerations

Our paper highlights the evaluation of LLMs on code-switching, a common phenomenon in the multilingual community. The research was carried out in compliance with the principles of academic integrity, including honesty, transparency, and rigor. The data used in this study was collected in accordance with ethical guidelines, and all participants provided informed consent. Within our study, we are aware of the potential impact that comes with our work and our experiments replicate prior work under comparable experimental conditions. We also ensured that the study did not cause harm or distress to any individuals or communities. The findings of this study have important implications for the development of multilingual LLMs and their potential applications in code-switching tasks. However, we acknowledge that further research is needed to address the limitations and gaps identified in this study. We believe that responsible and ethical use of language technology is crucial for creating just and equitable systems that benefit all individuals and communities.

## Acknowledgements

We would like to thank Xinyu Hua and Samson Tan for the constructive feedback and helpful discussion on our project.

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

## A  Fine-tuning Model Setup

We use a standard training setup for SA tasks: we fine-tune the models for a maximum of 15 epochs using the Adafactor (Shazeer and Stern, 2018) optimizer with a learning rate of 2e-5. All sequences are limited to a maximum sequence length of 256 tokens, truncating all sequences longer than this length, and dynamically padding the shorter sequences to the longest sequence length in their batch. All setups use a batch size of 128. We also use a linear warmup schedule, warming up for the first 10% of training steps before linearly decaying to 0. We measure Accuracy and Macro F1 as metrics for all setups, loading the best checkpoint based on the F1 score at the end for evaluation.

Word-Level LID setups use the same one as with NLU tasks, except we only train for 3 epochs and use a weight decay of 0.01. Given that one word may be split into multiple tokens during tokenization, we first realign the labels by setting the word label as the label of its first token, then setting the labels of all succeeding tokens as `-100`. This "dummy" label is then ignored during loss computation. We also load the best checkpoint and use the same metrics as with NLU, in addition to Precision and Recall.

For MT, we fine-tune for a maximum of 10 epochs using the Adafactor optimizer with a learning rate of 5e-5, loading the best checkpoint at the end. As the MT0 models are trained with instruction prompts, we also prepend a "prompt" to all sequences during fine-tuning in the form of `Translate [src] to [tgt]: [sequence]`. For the M2M100 and mBART models, we force the decoder's first token to be the language token of the target language. All setups use a batch size of 512 sequences. We also use a similar linear warmup schedule as with the SA and LID task setups. For MT, we use spBLEU as our performance metric and load the best model for evaluation based on it.

SUM follows most of the same setup that MT uses, except we only fine-tune for 3 epochs. For MT0, we use `Summarize: [sequence]` as our "prompt" that is prepended to all samples. We use ROUGE (ROUGE1, ROUGE2, ROUGEL, and ROUGEL-SUM) as our performance metric, loading the best model for evaluation based on it.

## B  Fine-tuning Model Results

| Machine Translation | | | |
|---|---|---|---|
| **Model** | **Size** | **BLEU** | |
| | | Hng→Eng | Eng→Hng |
| **Finetuning** | | | |
| M2M100 | 418M | 28.53 | 12.40 |
| M2M100 | 1.2B | 28.55 | 13.81 |
| mBART-50 | 611M | 25.50 | 12.10 |
| mBART-50$_{O2M}$ | 611M | 23.40 | 13.34 |
| mBART-50$_{M2M}$ | 611M | 29.53 | 13.38 |
| mT0 | 300M | 15.73 | 7.03 |
| mT0 | 580M | 24.97 | 11.88 |
| mT0 | 1.2B | 31.03 | 12.87 |
| mT0$_{prompted}$ | 300M | 16.66 | 7.24 |
| mT0$_{prompted}$ | 580M | 25.47 | 12.28 |
| mT0$_{prompted}$ | 1.2B | 31.88 | 13.90 |

Table 4: Results for all finetuned models for machine translation task.

## C  Prompt Templates

This section lists all prompts used for our experiment.

### Sentiment Analysis

- ```
  [INPUT] => Sentiment:
  ```
- ```
  Text: [INPUT] => Sentiment:
  ```
- ```
  [INPUT]
  What would be the sentiment of the text above?
  ```
- ```
  What is the sentiment of this text
  Text: [INPUT]
  Answer:
  ```
- ```
  Text: [INPUT]
  Please classify the sentiment of above text. Sentiment:
  ```

### Machine Translation

- ```
  Translate the following text from [SOURCE] to [TARGET].
  Text: [INPUT]
  Translation:
  ```
- ```
  [INPUT]
  Translate the text above from [SOURCE] to [TARGET].
  ```
- ```
  Text in [SOURCE]: [INPUT]
  How would you translate that in [TARGET]?
  ```
- ```
  Translate the following [SOURCE] text from to [TARGET].
  Text: [INPUT]
  Translation:
  ```
- ```
  Text in [SOURCE]: [INPUT]
  Text in [TARGET]:
  ```

[SOURCE] and [TARGET] are Hinglish and English.

### Summarization

- ```
  Summarize the following conversation in  English.
  Conversation: [INPUT]
  Summary:
  ```
- ```
  [INPUT]
  Summarize the above conversation in English:
  ```
- ```
  Conversation in [SOURCE]: [INPUT]
  How would you summarize that in English?
  ```
- ```
  Summarize the following [SOURCE] conversation.
  Text: [INPUT]
  English summary:
  ```
- ```
  Conversation in [SOURCE]: [INPUT]
  Summary in English:
  ```

[SOURCE] is either Hinglish or English.

**Word-level LID**

- ```
  Determine the language for each token in the text below with [ word | tag ].
  Use lang1 for [LANG1], lang2 for [LANG2], and other for others.
  [INPUT]
  ```

- ```
  For each token, identify its language (lang1: [LANG1], lang2: [LANG2], other) using [ word | tag ].
  [INPUT] =>
  ```

- ```
  Assign language tags to words: lang1 for [LANG1], lang2 for [LANG2], other otherwise.
  Format: [ word | tag ].
  [INPUT] =>
  ```

- ```
  [INPUT]
  Can you tag the language of each word in the sentence above: lang1 ([LANG1]), lang2 ([LANG2]), or
  other using format: [ word | tag ]?
  ```

- ```
  [INPUT]
  Label each word in the text above with its language: lang1 for [LANG1], lang2 for [LANG2], or other.
  Format: [ word | tag ].
  ```

[LANG1] and [LANG2] are English and Hindi for LID-Hindi-English data, and Modern Standard Arabic and Egyptian Arabic for LID Standard-Egyptian Arabic data.

# D  Detailed Results

Breakdown results of SA and LID across different languages can be seen in Figure 5 and Figure 6.

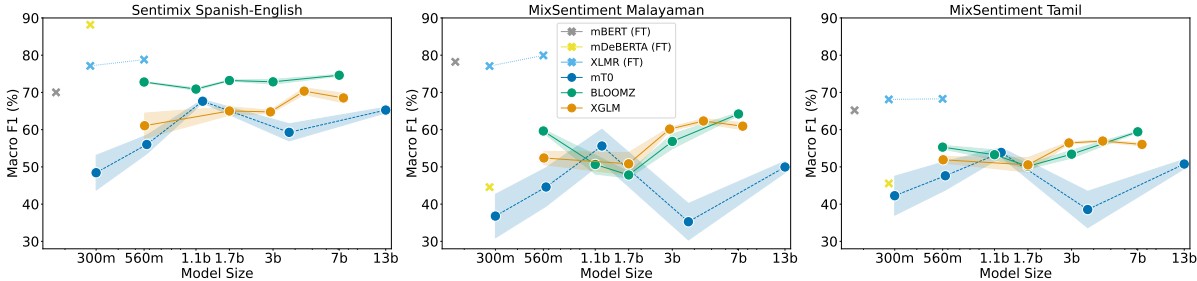

Figure 5: LLMs' sentiment analysis evaluation on **(left)** Sentimix Spanish-English, **(center)** MixSentiment Malayaman-English, and **(right)** MixSentiment Tamil-English.

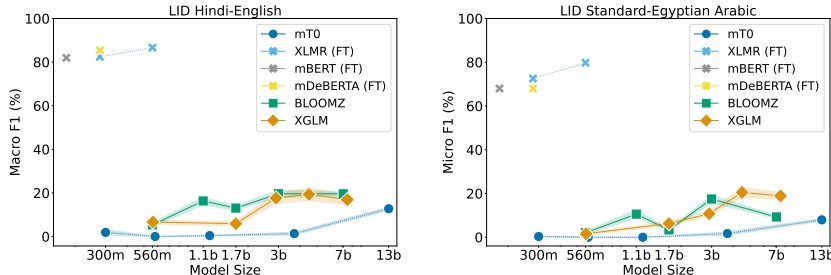

Figure 6: LLMs' word-level LID evaluation result on **(left)** Hindi-English word-level LID and **(right)** Standard-Egyptian Arabic word-level LID.