# OpenReview forum: "Multilingual Large Language Models Are Not (Yet) Code-Switchers"
_EMNLP/2023/Conference — EMNLP 2023 Main_

### Official Review · Reviewer_ajnS · 2023-08-01

**Soundness:** 4

**Excitement:**

4: Strong: This paper deepens the understanding of some phenomenon or lowers the barriers to an existing research direction.

**Paper Topic And Main Contributions:**

This position paper proposes to conduct empirical studies on the effectiveness of Multilingual Large Language Models (LLMs) in the domain of code-switching (CSW) and argues that current Multilingual LLMs are not yet capable of solving CSW tasks with satisfactory performances under zero-shot or few-shot settings.

The authors tested four types of Multilingual LLMs (mT0, BLOOMZ, XGLM, and OpenAI's GPT-3.5-turbo) with parameter sizes ranging from hundreds of millions to over 10 billions on four tasks: sentiment analysis, machine translation, summarization, and word-level language identification. Fine-tuning based methods are employed as baselines.

Analysis and observations on the results are provided for each of the four tasks, signaling the overall inferior performance of the examined LLMs to the fine-tuned methods in the domain of CSW. The authors also provided additional ad hoc analysis experiments (e.g., the monolingual setting on the summarization task and the k-shot experiments on the LID task).

Future research directions are also highlighted, mainly on three aspects: 1) better data representation for CSW in the multilingual pretraining  and/or instruction-tuning process, 2) adapting and extending existing CSW optimization objectives in relevant domains such as CSW speech processing to NLP, and 3) socially aware NLP technologies with CSW compatibility to foster inclusiveness across communities.

**Questions For The Authors:**

I have the following questions for the authors:

A) I wonder if there is a particular reason not to use the prompting methods / more recent open-source LLMs, is it budget, experiment time constraints, or experiment design related?

B) Regarding improving LLMs on domains, I notice you tuned smaller mT0 models. Did you explore tuning LLMs larger than 1.2B? Do you think parameter-efficient tuning (e.g., LoRA) can be effective for those models?

C) Referring to your first point in future research directions (fairer data representation), do you think LLM can be used towards the data augmentation in CSW?


**Reasons To Accept:**

In general, this paper is well motivated and empirically solid. The following strengths/reasons to accept can be identified:

- CSW can be a useful direction for inclusive and impactful NLP technologies. In the current time of LLM, position papers such as this can be beneficial to the community in identifying the upper limit of zero-/few-shot LLM inferences, and in understanding the gaps between the capabilities of existing LLMs and practical application needs.

- The holistic experiment results, applying a battery of LLMs on a diversity of tasks, convincingly support the authors' main argument of the weaknesses in LLMs' capabilities.

- The paper is well-written. The narrative and presenting logics of the empirical studies are easy to follow.

**Reasons To Reject:**

My concerns are mainly on whether the LLM capabilities have been fully unleashed. In my opinion the paper can be improved in the following aspects:

- The prompt templates used in the paper are still simple. Prompting methods such as Chain-of-Thought prompting (https://arxiv.org/abs/2201.11903) and Least-to-Most prompting (https://arxiv.org/abs/2205.10625) are proved effective in many use cases. In the case of CSW, these prompting methods might help decompose the CSW tasks into 1) recognizing the textual elements to be code-switched, and 2) organizing the elements of the different languages into proper outputs.

- The latest open-source LLMs such as the LLaMA based models are not included in the experiments. Therefore it remains unknown that whether stronger backbone LLMs will lead to a smaller or even eliminated performance gap.

**Reproducibility:**

4: Could mostly reproduce the results, but there may be some variation because of sample variance or minor variations in their interpretation of the protocol or method.

**Reviewer Confidence:**

4: Quite sure. I tried to check the important points carefully. It's unlikely, though conceivable, that I missed something that should affect my ratings.

---

> ### Author Rebuttal · Authors · 2023-08-29
>
> Thank you for the positive and constructive comments. We address your review as follows:
>
> Q: *Is there a particular reason not to use newer prompting methods and more recent open-source LLMs?*
>
> Great points on experimenting with other forms of prompting methods! To the best of our knowledge, these intricate prompt designs are generally effective in enhancing LLM's performance in reasoning-oriented tasks, whereas our project is primarily concentrated on linguistic aspects. However, we want to emphasize that the main contribution of the paper is to provide a benchmark performance of the models on various tasks. Models with these simple prompt templates we adopted are able to deliver outstanding performance in a non-CSW benchmark. Therefore, we hope to present these existing challenges without much engineering on prompt specificity and leave this for future research. We appreciate these references and considerations and will include them in our revised version.
>
> For the reason why more recent LLMs are omitted: we chose our benchmark models among LLMs that 1) presented or advertised themselves as multilingual, and 2) had in their pretraining data all the languages that we were testing on.
> While LLaMA v1 certainly outperforms other competitive LLMs during the time of writing the paper, it is mostly trained on monolingual English data and it is not outright suggested for multilingual use, nor did its pretraining data contain Malayalam, Tamil, or Hindi. This is similar to LLama 2 and many other LLMs and, therefore excluded from our benchmark. We will include this selection criteria in our revised paper.
>
> Regarding the inclusion of ChatGPT in the benchmark, we decided to include it because, at the time of writing, it was considered the “best performing” closed-source model. We believe that creating a benchmark of open-source vs. closed-source models is important for future reference.
>
> Q: *Did you explore tuning LLMs larger than 1.2B? Do you think parameter-efficient tuning can be effective for those models?*
>
> The omission of full finetuning on models larger than 1.2B is largely due to a constraint on time and computation resources.
>
> While we believe that parameter-efficient methods such as LoRA will make finetuning the larger models feasible, these methods will incur some form of performance loss compared to full finetuning. This is acceptable in most applications, but in this study, finetuning was used as a comparative benchmark, and so we believe that full finetuning is needed to properly represent their true performance.
>
> However, considering that smaller finetuned models already outperform zero-shot model for code-switched tasks and that the scaling law for finetuned models is consistent, we believe it is safe to hypothesize that finetuning models larger than 1.2B will result in performance stronger than smaller models and far stronger than prompted models.
>
> We do, however, agree that it is also good to get benchmark performance on the larger variants of models for full finetuning in the future. Stronger performance will further empirically confirm our conclusions, while weaker performance may yield insights into problems in pretraining techniques, quantity of data, distribution of languages, etc when done at scale.
>
> Q: *Can LLMs be used for data augmentation for code-switched languages in the future?*
>
> At the moment, as shown by our results and other corroborating research (https://arxiv.org/abs/2303.13592), we do not advise using prompting methods for producing code-switched text. Finetuned models, on the other hand, maybe more amenable to code-switched data augmentation with current techniques (such as back translation, which is already widely used in machine translation).
>
> All in all, our findings further show the need for techniques and datasets to be made to better support code-switching in the future.

---

### Official Review · Reviewer_dySR · 2023-08-03

**Soundness:** 4

**Excitement:**

4: Strong: This paper deepens the understanding of some phenomenon or lowers the barriers to an existing research direction.

**Paper Topic And Main Contributions:**

This paper investigates how pre-trained fine-tuned language models and large language models perform on natural language tasks where the input includes code-switched texts. Code-switching (CSW) refers to the use of multiple languages within a single utterance, facilitating, for example, communication of culture-specific concepts. The paper considers a wide variety of language models, including the well-known ChatGPT, and four NLP tasks: sentiment analysis, machine translation, text summarization, and word-level language identification. The authors present and analyze the results of applying the models to CSW datasets, finding that pre-trained multilingual models with fine-tuning still outperform most LLMs by a wide margin. They make suggestions for making LLMs more robust against CSW, and argue for the benefits of doing so.

**Reasons To Accept:**

This paper presents a thorough investigation into the abilities of LLMs to handle code-switched text. The experimental design is robust: many different models, including pre-trained LMs such as mBERT, XLMR, and LLMs such as BLOOMZ and GPT, are tested; various model sizes are considered, where applicable, and parameter counts are carefully tracked and plotted against performance.

The tasks are also well chosen, with classification, tagging, and generation tasks all represented. Different language mixes are also considered.

The presentation of the results is clear, with the plots of performance against parameter count being particularly welcome.

Finally, the paper is overall clear and well-written, with a clear focus, a well-motivated problem, and empirically sound conclusions.

**Reasons To Reject:**

The paper's greatest weakness is shallow analysis of the model outputs and results. While it is clear how the different models compare, it is not made clear why. What kinds of errors to LLMs make that fine-tuned pre-trained models do not? Are the errors made by smaller versions of a model a superset of the errors made by larger versions? GPT does much better than all other LLMs on language identification by an incredible margin -- this should surely be explored further. References to specific examples, showing the different errors made or avoided by different models would help answer these questions and greatly strengthen the paper.

It would also be helpful to have some reference to how the models perform on comparable non-CSW datasets. This would help to contextualize the results, and to differentiate between models that struggle with CSW, and models that struggle on particular languages. For example, GPT performs relatively well on Hng-Eng translation, compared to other LLMs. Is that because it is more robust against Hng CSW, or because it is simply more robust on Hindi-English translation, and that transfers to Hng-Eng? Even though GPT cannot be analyzed directly, such information could help us to understand the nature of its advantage over other LLMs.

Finally, it would be good to discuss the nature and quantity of the training data available to each model, and how that may impact its performance. At one point, the authors note that they "suspect that models have seen these tasks in a monolingual fashion during pretraining" -- is it possible to verify this for any of the models tested? Similarly, just as the authors report and discuss the number of parameters each model has, would it also be possible to report the quantity of text (in the relevant languages) that each model was exposed to? Given the proprietary nature of some models, this may not be possible (or may be possible only to a limited extent), but if this is the case, the authors should discuss this problem explicitly.

**Reproducibility:**

3: Could reproduce the results with some difficulty. The settings of parameters are underspecified or subjectively determined; the training/evaluation data are not widely available.

**Reviewer Confidence:**

4: Quite sure. I tried to check the important points carefully. It's unlikely, though conceivable, that I missed something that should affect my ratings.

---

> ### Author Rebuttal · Authors · 2023-08-29
>
> Thank you for your comment. We address your comment as follows:
>
> Q: *“Shallow analysis of the model outputs and results”*
>
> A: We appreciate your valuable suggestions on in-depth performance analysis and adding specific examples.
> We found a lot of interesting qualitative patterns that are challenging to draw into a single conclusion. We will add some of our observations in camera-ready.
> * For sentiment analysis, models of different sizes do not show a strict scaling law pattern and also errors of smaller ones are not supersets of those of larger ones.
> * In machine translation, in a five-shot setting, mT0-13B could only generate English instead of Hinglish as prompted. For example, with a prefix of 5 other examples, we asked the model “Translate the following text from English to Hinglish. Text: hello there, I have not seen this movie so im going to take a minute to look it over :) Translation:”, it generates “hello there, I have not seen this movie so I going to take time to look it over:).” when the ground truth should be “hello yar, mein is movie ko nahi dekha hoon tho, tho mein thode der ke liye isko dekh loonga”  BLOOMZ also suffers from a similar issue.
> * For language identification, we’ve mentioned GPT’s outstanding performance at the beginning of section 3 and also section 3.6. The main reason why LLMs do not perform well is that they are unable to generate sequences in the format requested in the prompt. Still, in a five-shot setting, we asked BLOOMZ-7B1 to label the sentence “we the fans luv you , sirji”. The model wrongly generates “[ we the fans | lang1 ] [ you | lang1 ] [ sirji | lang1 ] [, | other ]”  because it’s asked to put words individually in the brackets as well as missing some words from the original sentence. Our hypothesis for the large performance gap between GPT and other LLMs in following instructions is the RLHF and instruction tuning in GPT’s training process.
>
> We will include a more in-depth qualitative analysis with these specific examples in the revised paper.
>
>
> Q: *It would also be helpful to have some reference to how the models perform on comparable non-CSW datasets.*
>
> A:  In Figure 3 and Table 3, we compared model performances on Hindi->English MT and English -> English summarization tasks against their Hinglish->English direction, showing that models generally do better in pure monolingual settings. Also thanks for noticing GPT's performance in the MT task! Combining the results of MT and summarization, GPT does much better in MT in Hinglish → English than other models but was not the best-performing one in summarization of the same language. We are unfortunately hindered by GPT’s proprietary nature to fully explore its robustness between monolingual and code-mixed settings, but we will extend our comparison in Table 3 to GPT3 in the revised paper for a more comprehensive analysis.
>
>
> Q: *"suspect that models have seen these tasks in a monolingual fashion during pretraining" -- is it possible to verify this for any of the models tested?*
>
> A: Great suggestion on “report and discuss the number of parameters for each model”! We observed that while mT0 and BLOOMZ have a higher percentage of Tamil in their pretraining data compared to Malayalam, their performance varies by language in the sentiment analysis task, each showing different strengths. We also noticed that in the pretraining data for XGLM, the quantity of Arabic is nearly twice as much as Hindi. Despite this, when it comes to performance in the Language Identification (LID) task with a mix of English, both languages show similar results.  We will include language percentage and more discussion on our observations above in the revised version.
>
> We have verified that all the languages included in our study are in the pretraining corpus of either the finetuning models or multilingual large language models. And we want to emphasize the contribution of this study lies in assessing the capabilities of currently available multilingual LLMs. A more controlled way to observe the effect of language factors is by training models in a leave-one-out setting. However, given time and budget constraints, we will leave this for future research.

---

### Official Review · Reviewer_RRwZ · 2023-08-11

**Soundness:** 4

**Excitement:**

3: Ambivalent: It has merits (e.g., it reports state-of-the-art results, the idea is nice), but there are key weaknesses (e.g., it describes incremental work), and it can significantly benefit from another round of revision. However, I won't object to accepting it if my co-reviewers champion it.

**Paper Topic And Main Contributions:**

The paper evaluated the capabilities of multilingual large language models (LLMs) on code-switched (CSW) data by answering the question "Can existing multilingual LLMs effectively understand code-switching?". To answer this question, they evaluated several LLMs on 2 major downstream tasks which are language generation and sequence classification, and in addition compared their performance against fully fine-tuned smaller pretrained LMs.Their analysis shows that these LLMs underperform at the different tasks when compared to the performance of the fine-tuned PLMs.

**Questions For The Authors:**

1. Where are there no Chat GPT 3.5. results on Figure 2? GPT 3.5 was mentioned under Overall Results in section 3.
2. How many examples did you use to fine-tune the PLMs?
3. Figure 2: What is mBART-50 M2M (FT)?

**Reasons To Accept:**

1. The problem was well motivated
2. This is a work in low-resource setting due to scarcity of CSW data and these findings would be beneficial for low-resource languages. They evaluated numerous LLMs, testing their different sizes too.
4. They evaluated the LLMs on different number of examples, 5 different prompt templates, and this was carried out on 4 tasks.
5. They highlighted major implications of their findings which would be usefully for the field

**Reasons To Reject:**

1. it seems that there is no fair comparison between the LLMs and the finetuned PLMs. The paper never mentioned the number of training samples that were used for fully-fine tuning the PLMs
2. It is also not clear if these results are generalizable due to the few languages evaluated on. For instances, the LLMs were only evaluated for machine translation and summarization using just Hinglish-English data.

**Reproducibility:**

4: Could mostly reproduce the results, but there may be some variation because of sample variance or minor variations in their interpretation of the protocol or method.

**Reviewer Confidence:**

4: Quite sure. I tried to check the important points carefully. It's unlikely, though conceivable, that I missed something that should affect my ratings.

**Typos Grammar Style And Presentation Improvements:**

- Line 638 & 644 repeated citation
- Line 652 & 656 repeated citation
- No legends in Figure 5

---

> ### Author Rebuttal · Authors · 2023-08-29
>
> Thank you for your insightful review! We address your comments as follow:
>
> (1) Thank you for your suggestion on including detailed statistics of training samples! For sentiment analysis, we have 2,571, 8,831, 9,075 samples for Mal-Eng, Spa-Eng and Tam-Eng respectively. For machine translation, we have 8,060 pairs of hinglish-english sentences. For summarization, we have 5,831 source-target pairs. For language identification, we have 4,823 and 8,464 training samples for Hin-Eng and MSA-EA respectively. We will include these in our revised paper. Regarding “fair comparison”, though the number of samples used for finetuning is much more than that of the examples used for few-shot learning, one of the main objectives of the paper is to compare between finetuning smaller models and zero-shot/in-context learning capabilities of LLMs and investigate whether their vast amount of multilingual pretraining data contributes to code-switched tasks. These LLMs are restricted by their max input length, and our input lengths also vary depending on different task formulations. Therefore we controlled the number of shots to be consistent among different tasks. In a “fairer” setting, we could increase the number of shots for LLMs but our study shows that more shots do not help performances, and we leave this for future research.
>
> (2) The number of languages is mainly limited due to the severe lack of publicly available code-switching language resources. The highly oral nature of the code-switching practice presents additional difficulties in gathering and annotating data. For MT and summarization, the hinglish resources we included in our study are the only ones that can be easily accessed online with a permissive license. Datasets without a clear license have been filtered out. Our benchmark highlights the limitations of existing LLMs when dealing with code-switched data. We hope this will encourage more interest within the community in collecting high-quality code-switching data and related research topics. We also remain hopeful and are happy to include new languages with suitable resources.
>
>
> Q1: *Why are there no ChatGPT (GPT-3.5-Turbo) results on Figure 2?*
>
> Thank you for spotting that! Given the closeness of the model, we are uncertain of the size of GPT-3.5-turbo, therefore, did not include it in Figure 2. Please refer to Table 2 for a more comprehensive view of the performances of all model variants.
>
>
> Q2:  *How many examples did you use to fine-tune the PLMs?*
>
> We used the entire training datasets for the benchmark tasks that we used. Please see detailed statistics in (1). We will also include those in our revised version of the paper.
>
>
> Q3. *What is mBART-50-M2M (FT)?*
>
> This refers to the variant of the mBART-50 model that was finetuned (FT) for many-to-many translation (M2M) as released by Meta AI (https://arxiv.org/pdf/2008.00401.pdf).
>
>
> Thanks for spotting the duplicate citations! We will correct them in the revised paper. For Figure 5, the legends are the same for the subfigures and therefore it’s in the middle one.

---

### Meta-Review · Area_Chair_P6nc · 2023-09-18

**Recommendation:** 5

**Metareview:**

This paper evaluated the capabilities of multilingual LLMs on code-switched (CSW) data based on direct fine-tuning and prompting of LLMs. The evaluation is performed on four downstream tasks (sentiment analysis, machine translation, summarisation and word-level language identification) across four code-switched English corpus (Spanish-English, Malayalam-English, Tamil-English, Hindi-English/Hinglish, and MS Arabic-Egyptian Arabic).

All the reviewers see the evaluation of LLMs on CSW data has a promising direction especially evaluation in low-resource setting and a well motivated problem. The selection of tasks and models are also adequate to prove the point in general about the inability of LLMs to perform well on CSW data. There a few concerns raised by the reviewers like providing fine-tuning data sizes, non-evaluation of more recent LLMs like ChatGPT, no qualitative analysis of model output, comparison with non-CSW data, many of these concerns have been appropriately answered by the authors.

---

### Decision · Program_Chairs · 2023-10-07

**Decision:**

Accept-Main

**Comment:**

This paper evaluated the capabilities of multilingual LLMs on code-switched (CSW) data based on direct fine-tuning and prompting of LLMs. The evaluation is performed on four downstream tasks (sentiment analysis, machine translation, summarisation and word-level language identification) across four code-switched English corpus (Spanish-English, Malayalam-English, Tamil-English, Hindi-English/Hinglish, and MS Arabic-Egyptian Arabic).

All the reviewers see the evaluation of LLMs on CSW data has a promising direction especially evaluation in low-resource setting and a well motivated problem. The selection of tasks and models are also adequate to prove the point in general about the inability of LLMs to perform well on CSW data. There a few concerns raised by the reviewers like providing fine-tuning data sizes, non-evaluation of more recent LLMs like ChatGPT, no qualitative analysis of model output, comparison with non-CSW data, many of these concerns have been appropriately answered by the authors.